# A Study on Water-Induced Damage Severity on Diesel Engine Injection System Using Emulsified Diesel Fuels

Min-Seop Kim, Ugochukwu Ejike Akpudo and Jang-Wook Hur *

Department of Mechanical Engineering (Department of Aeronautics, Mechanical and Electronic Convergence Engineering), Kumoh National Institute of Technology, 61 Daehak-ro (yangho-dong), Gumi 39177, Gyeongbuk, Korea; alstjq1104@kumoh.ac.kr (M.-S.K.); akpudougo@gmail.com (U.E.A.)
* Correspondence: hhjw88@kumoh.ac.kr

**Abstract:** Diesel engine emissions contribute nearly 30% of greenhouse effects and diverse health and environmental problems. Amidst these problems, it is estimated that there will be a 75% increase in energy demand for transportation by 2040, of which diesel fuel constitutes a major source of energy for transportation. Being a major source of air pollution, efforts are currently being made to curb the pollution spread. The use of water-in-diesel (W/D)-emulsified fuels comes as a readily available (and cost-effective) option with other benefits including engine thermal efficiency, reduced costs, and NOx reduction; nonetheless, the inherent effects—power loss, component wear, corrosion, etc. still pose strong concerns. This study investigates the behavior and damage severity of a common rail (CR) diesel fuel injection system using exploratory and statistical methods under different W/D emulsion conditions and engine speeds. Results reveal that the effect of W/D emulsion fuels on engine operating conditions are reflected in the CR, which provides a reliable avenue for condition monitoring. Also, the effect of W/D emulsion on injection system components-piston, nozzle needle, and ball seat–are presented alongside related discussions.

**Keywords:** common rail diesel engine; water-emulsified fuel; damage severity; diesel injection system; signal processing

## 1. Introduction

Across different industries, energy is mostly harvested from diesel engines due to the comparative thermal, maintenance, and cost efficiencies associated with them [1,2]. Contrary to these benefits, diesel engine emissions constitute a significant portion of environmental pollution since these emissions contain carbon monoxide (CO), nitrogen oxides (NOx), and particulate matter (PM)—the chief sources of health and environment-related issues. Studies reveal that amidst the revelation that diesel engine emissions contribute nearly 30% of greenhouse effects, it is estimated that there will be a 75% increase in energy demand for transportation by 2040, of which diesel fuel constitutes a major source of energy for transportation. This also hints at an expected 85% increase in diesel fuels for heavy-duty applications [3].

The injection system is a primary unit of a functional CR diesel (CRD) engine, which typically consists of a pumping system, a CR system, and combustion chamber(s) [4]. Ideally, the injection system functions properly with uncontaminated diesel fuel and high-quality component parts; however, the reality is that most diesel fuels are not as pure as we believe. This is one of the major reasons that periodic engine check-ups, filter and oil replacements, and fuel quality monitoring are recommended [5]. Fuel composition largely determines the efficiency and reliability of the engine; however, in recent years, the price of volatile fuels and strict emission regulations gave considerable impetus to the study of emulsion diesel fuels of various configurations. Quite unconventionally, this quest to improve fuel quality, reduce emission severity, and minimize costs motivated water emulsification of diesel fuels. Accordingly, these factors also motivated several studies

that mostly report experimental investigations of engine performance and exhaust gas emissions under operating conditions of W/D emulsion fuels [6–9].

Driven by the increasing demand for emissions reduction, recent developments in the automotive industry are channeled towards achieving higher power densities for mass-produced automotive diesel engines [10,11]. These advancements in designs for diesel engine are mostly geared towards improving the air charging system, achieving lower compression ratio combustion systems, and most significantly, modeling/developing diesel fuel injection system (FIS) designs capable of 2700 bar and above. Consequently, concepts like engine downsizing (in an attempt to minimize engine emissions while achieving high efficiencies) and operating parameter optimization/control for higher FIS outputs are being explored [12–14]. For instance, the FIS design recipe for a 100 KW/L engine power density provided in [12] is expected to provide an injection pressure as high as 2700 bar. While their simulated investigations on a high-power design- ultralight midsize diesel engine matches future $CO_2$ targets [14], concerns for real-life applicability are still high, and with the already-increasing demand to reduce (rather than maintain) engine emission levels on a global scale, additional concerns on the wear effects of W/D emulsion fuels in such high-power designs are still open for further investigation.

Against the cost-effectiveness of diesel fuel emulsification, concerns about the right W/D ratio, and more importantly, the inherent engine power loss, component corrosion, seal damage, and other injection system problems associated with the practice, still remain [15–17]. Although some may argue otherwise, the compelling yet contradictory arguments only further motivated this study. Consequently, this study analyzes the effect of W/D emulsion fuels on a passenger car—KIA Sorento 2004's engine as a case study—and makes the following contributions:

- By exploiting the rail pressure sensor (RPS) measurements from a passenger car's diesel engine, the effect of the effect of W/D emulsion fuels on the FIS was analyzed. This was done by investigating the spectral response of the signals over different W/D emulsions and varying engine speeds.
- The effect and damage severity of diesel emulsification on the injection system components were investigated and presented. A degradation/wear assessment was conducted for the injectors and necessary judgments/inference recorded for possible root causes and correlations between fuel conditions and engine performance.
- Extensive empirical and descriptive deductions are presented. The findings are expected to provide a reliable paradigm worth considering for CRD engine condition monitoring, failure diagnostics, improved engine design, and decision making.

## 2. Literature Review on Related Works

Recently, a research team conducted various experiments for finding a better alternative to diesel used in conventional CRD engines [18]. In their experiments, different W/D emulsions were prepared for comparing emission performance, combustion parameters and for finding a better replacement for the diesel used in conventional engines. Their findings reveal that the emission parameters were diminished with eucalyptus in water emulsion when correlated with W/D emulsion, while overall, the best result was provided at 15% water content. The study concluded that W/D emulsion proves to be a better fuel than conventional diesel. Being a recent study, their findings not only hints at the superiority of emulsified diesel fuels over the conventional diesels, but they also provide strong paradigms for making necessary considerations (and criticisms) for a global adoption. In support of their findings, another research team who conducted a similar study compared the 10% W/D emulsion with pure diesel fuel to evaluate the effect on engine performance and exhaust gas, and conducted experiments under various engine rotation speed conditions. Although the output and engine efficiency were lower than that of pure diesel fuel, it produced an engine efficiency similar to that of pure diesel fuel at high engine speed, and brake specific fuel consumption (BSFC) was higher in W/D emulsion at all rotation speeds. In addition, a decrease in exhaust gas temperature was also

observed [19]. Another similar study was applied to a diesel engine used in ships, which also reported efficient exhaust gas emission under W/D emulsion conditions [20]. As such, the mixed use of water and diesel has advantages such as controlling engine overheating, fuel efficiency, NOx and CO reduction, and operational cost reduction. Notwithstanding the significant benefits, the high oxidation potential of water on metal components, W/D emulsion adoption still poses strong concerns [21].

Against the advantages recorded in these studies, fewer attempts were made to study the water-induced damage to CRD engine injection system components. Wróblewski et al. [22] conducted an investigation on a KIPOR KDE3500E generator set engine with W/D emulsion fuels. Their observation on the atomizer revealed not only that water addition reduced NOx concentration in the exhaust gas by 30% (with 41% addition of water), it also revealed that addition of water over a period of 120 min does not necessarily contribute to injector nozzle wear. On a different note, the authors of [23] assessed combustion and emission behavior of a diesel engine by injecting water into the exhaust manifold. By opening the exhaust valve during the intake stroke, the injected water and exhaust gases reentered the engine cylinder, then mixed during the intake and compression stroke to participate in the combustion process. Results from this study showed that soot emissions reduced by up to 40% while NOx emissions reduced by 85% when compared to that of conventional diesel combustion. In a retrospective review, Vellaiyan and Amirthagadeswaran [24] discussed recent studies on the efforts geared towards research and development of W/D emulsion fuels for NOx, CO, and PM reduction alongside emulsion fuel stability, physio-chemical properties, and effect of W/D emulsion fuel on combustion, performance, and emission characteristics. Like most other articles that focus on the advantages of W/D emulsion, this comprehensive study further discussed the role of nanoadditives in W/D emulsion fuel, nanofuel synthesis, and its influence on engine performance and emission characteristics; however, the long–term effects of using W/D emulsion fuels (particularly on the injection system components) are not yet fully explored.

In addition to analyzing the effect of W/D emulsion fuels on CRD FIS, this study provides empirical analyses of the FIS's CR dynamics for gaining reliable intuition on the FIS's operational performance under different W/D emulsion compositions. The case study presented herein involves a condition monitoring process using spectral parameters for determining a quantitative reference for failure mode and effect analysis (FMEA) on the engine. The rest of the study is organized thus: Section 2 presents the theoretical background of the spectral phenomena employed in this study, while Section 3 presents the materials and methods employed for proposed case study. Section 4 discusses the results from exploratory investigations, while Section 5 concludes the paper.

## 3. Theoretical Background

### 3.1. Common Rail Injection System

A typical CRD engine's FIS consists of the major parts—high-pressure pump (HPP), CR system, fuel injector, and the electronic control unit (ECU), as illustrated in Figure 1.

From the blue lines, the injection process commences with fuel delivery from the reservoir via the low-pressure pump to the HPP, which pumps the fuel through the red line to the CR for fuel distribution to the injectors. The ECU controls the opening and closing of the injectors and the pressure control valve (PCV) for maintaining acceptable fuel flow throughout the injection system. These pressure fluctuations in the CR system are reflected in the RPS, which provides reliable condition monitoring parameters that can be harnessed digitally. Also, the RPS signals reflect directly (in amplitudes and frequency components), the CR pressure dynamics as engine speeds varies. As the acceleration pedal is pressed down, both the amplitudes and and variance in the RPS signals are increased until the maximum engine speed is reached. Properly harnessing the spectral and transient information underlying in the signals over different W/D emulsions and engine speeds can provide a reliable perspective for making appropriate empirical inferences towards determining quantitative reference values for FMEA on the engine.

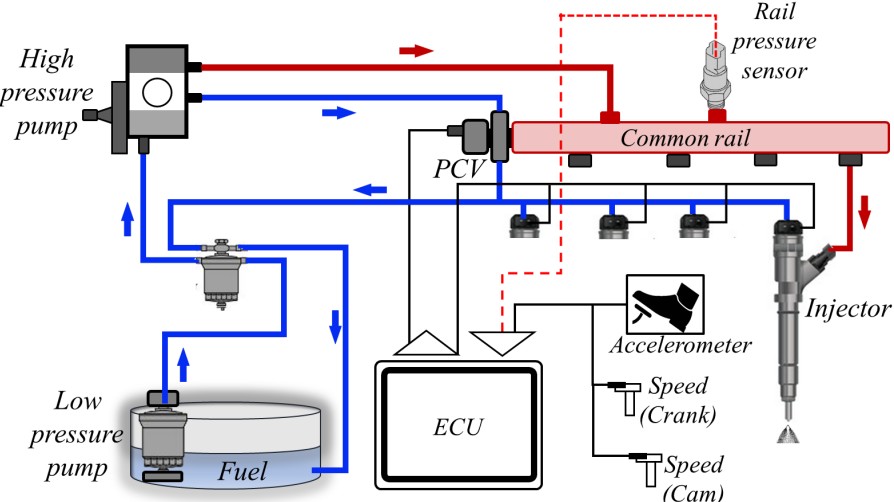

**Figure 1.** A typical common rail diesel fuel injection system.

### 3.2. Spectral Condition Monitoring

For decades, digital signal processing (DSP) was a reliable condition monitoring paradigm in diverse applications. Against the limitations of time-domain methods which reveal transient characteristics in a signal, the invention of various frequency-domain (and time-frequency-domain) techniques further provided even more reliable paradigms for stationary and nonstationary signal processing. Named after Joseph Fourier (21 March 1768–16 May 1830), the Fourier transform (and its variants) forms the basis for most frequency-domain signal processing techniques. The fluctuations in the CR system under different W/D emulsion conditions (and engine speeds) are reflected in the RPS measurements by different amplitudes and frequencies. Merely observing the signal amplitudes over time is not as reliable as observing the distribution of the signals over their constituent frequencies [25]. As a tool for investigation, the fast Fourier transform (FFT) and power spectral density (PSD) provide a condition monitoring avenue for analyzing the spectral behavior of the CR system. From them, different spectral parameters can be extracted for making necessary comparative inferences—-energy distribution, waveform, and other characteristics on the engine's FIS.

Under relaxable assumptions, most signals are composed of complex synthesis of Sine and Cosine functions, and this provides avenue for the FFT to flourish reliably. Although it lacks transient information and can only be extracted over a complete signal duration, given a time-record (one-dimensional sensor signal) $f(x) = \{x_1, x_2, \ldots, x_m\}$, the Fourier transform of a function $f(x)$ is traditionally denoted $F(k)$ and is computed using Equation (1) below:

$$F(k) = \int_{-\infty}^{\infty} f(x)e^{-j\left(\frac{2\pi mk}{N}\right)}dx, 0 \le m \le N \tag{1}$$

where $f(x)$ is the input signal, $k$ is the length of the transform, and $F(k)$ is the corresponding frequency-domain output of the signal.

By identifying the respective frequency components in the CR signals across different W/D emulsion compositions, and with the knowledge of the CR dynamics under conventional diesel, fault/anomaly detection can be achieved form a visual inspection of frequency spikes and statistical feature extraction from the frequency domain. In practice, the FFT is quite sensitive and outputs virtually all (including the insignificant) frequencies that constitutes a signal, and this high sensitivity sometimes limits its efficiencies for accurate condition monitoring. On the bright side, the PSD computes the energy densities of the constituent frequencies, thereby exaggerating relevance for high-energy signal components while suppressing the effects of the lower-energy constituents. PSD creates a

spectrum by outputting the magnitude squared of the FFT outputs from Equation (1) using Equation (2) below:

$$PS(k) = |F(k)|^2 \tag{2}$$

## 4. Materials and Methods

Figure 2a shows the passenger car—KIA Sorento 2004 model which was used as a case study—and Figure 2b illustrates the data acquisition process for the experiment. The engine specification is summarized in Table 1.

**(a)**                                                         **(b)**

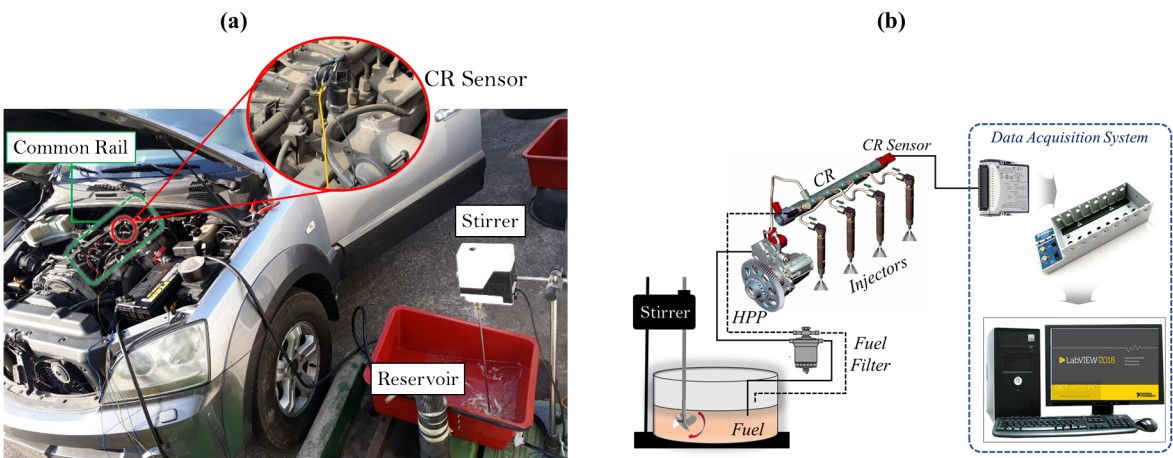

**Figure 2.** A view of (**a**) actual experiment setup; (**b**) schematic view of experimental setup.

**Table 1.** Test engine specifications.

| Car Model | Engine Type | Bore × Stroke (mm) | Maximum Power | Maximum Torque (Nm/RPM) | Compression Ratio | Fuel Injection | Aspiration |
|---|---|---|---|---|---|---|---|
| KIA Sorento 2004 | In-line, Four (4) | 91 × 96 | 138 hp @3800 RPM | 343 Nm @ 1900 RPM | 17.6 | Common Rail | Turbocharged, inter-cooled |

As shown in Figure 2b, the external reservoir was specially designed to accommodate an overhead stirrer (OSA-10 made by LK LABKOREA) for continued blending/mixing and to avoid phase separation for the W/D emulsion fuels. Then, the emulsions were respectively tested for stability with a centrifuge. Table 2 shows the different viscosity values of the emulsion compositions at 10 °C, whereby the emulsion compositions are represented as $EM - x\%$ (where $x$ represent the concentration of water by volume in $(100 - x)\%$ of diesel).

For each of these W/D emulsion compositions, CR pressure signals were digitally collected at varying engine speeds at 200 Hz sampling rate via the engine's RPS (Bosch 0 281 002 405) using a NI 9228 module (connected to a NI cDAQ 9178 and LabView software) produced by National Instruments CORP. The RPS design specification outputs a maximum of $(5 \pm 0.25)$ V at 2200 bar with a response time $(\tau_{10/90}) \leq 5$ ms and a maximum overpressure of 1800 bar (rupture pressure = 3000 bar). The engine exhaust temperature was also collected via a NI 9214 module at 1 KHz with an RTD thermocouple attached to the exhaust manifold. During data collection for the respective W/D emulsions, the injection system is flushed using the emulsion composition of interest for about an hour before the signals of interest are collected.

As shown in Table 2, a positive correlation is observed between the increasing W/D emulsions and their respective viscosity values, which suggests a possible positive correlation between increasing emulsion compositions and increased engine stress during combustion since a higher fuel viscosity demands higher engine power for a complete combustion.

**Table 2.** Viscosity of different W/D emulsion compositions.

| Emulsion Composition | EM-0% | EM-1.3% | EM-1.5% | EM-2.0% | EM-5.0% | EM-10.0% | EM-20.0% |
|---|---|---|---|---|---|---|---|
| Kinematic Viscosity (cSt) at 10 °C | 6.24 | 9.478 | 9.49 | 9.52 | 9.71 | 10.01 | 10.61 |

## 5. Experimental Results and Discussions

At the end of the experiment, the exhaust temperatures under the different W/D emulsions were analyzed for different engine speeds. Figure 3 shows the temperatures for EM-2.0% (in green), EM-5.0% (in blue), EM-10.0% (in red), and EM-20.0% (in black) at the increasing engine speeds—1200, 1500, 1700, and 2000 RPMs. Due to sensor line malfunction in the temperature module, the exhaust temperatures for EM-0%, EM-1.3%, and EM-1.5% were not recorded.

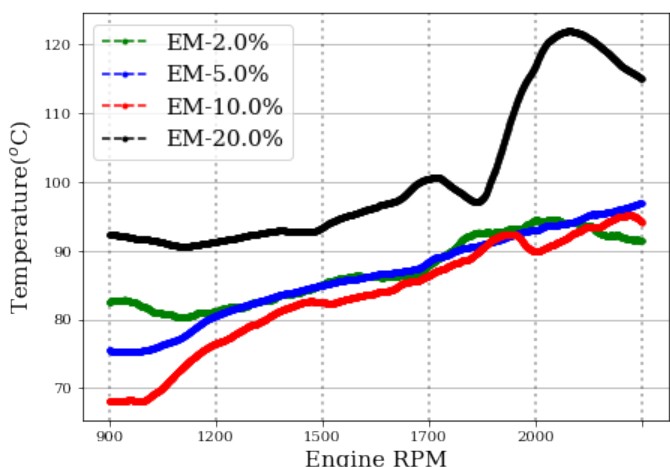

**Figure 3.** Exhaust temperature for different W/D emulsion compositions at increasing engine speeds.

At slower engine speeds (900–1200 RPM) between the green (EM-2.0%), blue (EM-5.0%), and red (EM-10.0%) lines, there is a clear cooling effect (reduction in temperature) resulting from increasing the emulsion compositions. For EM-20.0% (in black), the authors believe that the high W/D composition and the already existing FIS fault caused by EM-10.0% at higher engine speeds possibly induced an increased mechanical stress in the engine. This invariably resulted in overheating in compensation for the poor combustion process emanating from the *excess* water percentage. Notwithstanding, it is observed from the overall temperature trends that exhaust temperature is increased by increasing engine speeds.

### 5.1. Injection System Failure Investigation

The injector plays a role of injecting the fuel stored at high pressure in the CR into the combustion chamber at the exact time decided by the ECU. Ideally, a CRD engine injector is illustrated in Figure 4 which identifies the key constituent components.

The spray rhythms in the injectors are controlled by the engine's ECU via a solenoid valve (ball type) as shown in Figure 4. This valve operates thus: when an electric signal is received from the ECU, the valve spring is compressed to open the valve. Accordingly, the hydraulic pressure acting on the plunger drops while the nozzle needle valve opens as the hydraulic pressure becomes weaker than the force acting on the nozzle needle. This oscillatory movement of the plunger relies on very minimal friction between the plunger and the valve, which under realistic conditions is ensured by the lubricating effect and low viscosity of diesel. Unfortunately, the use of W/D emulsions leads to increased viscosity in the emulsion fuel (as verified in Table 2) and the oxidation of the metal contact surfaces. This invariably increases friction between the components, resulting in component wear in the long term.

To investigate for the injector efficiency, a back-leak test was performed on the CR and injectors and is shown in Figure 5.

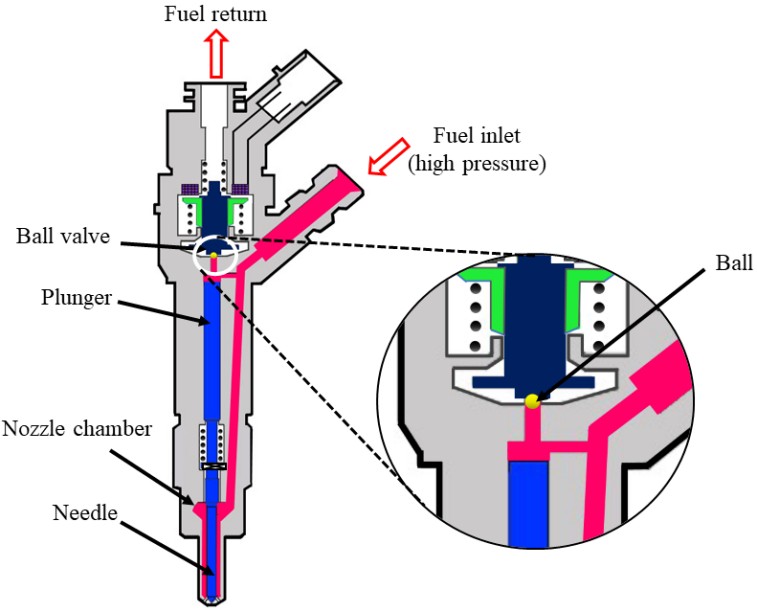

**Figure 4.** Lateral view of a CRD engine injector [26].

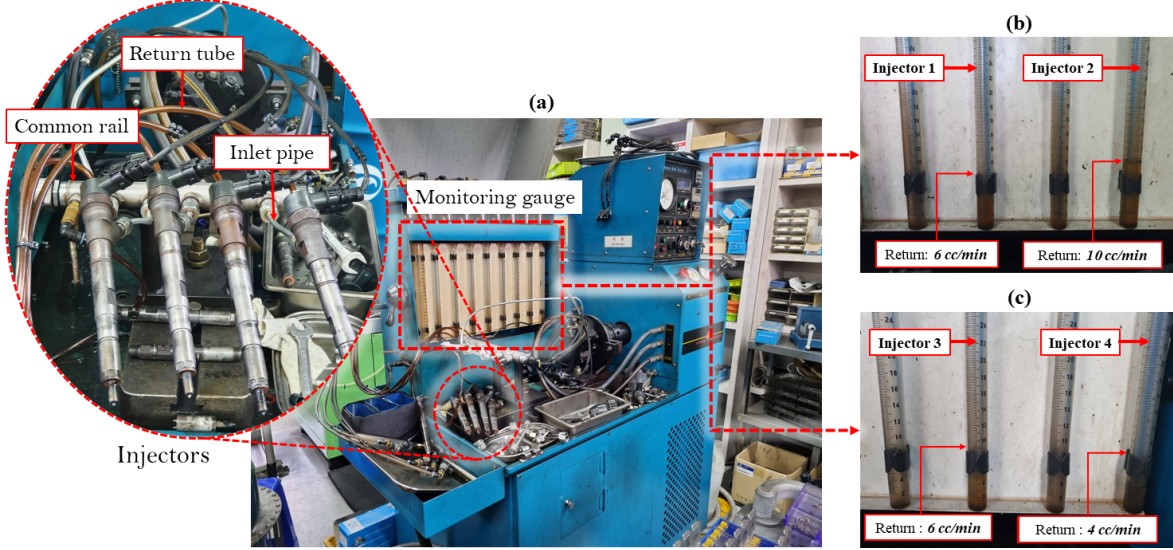

**Figure 5.** Back-leak test apparatus (**a**) and results (**b**,**c**).

Figure 5a shows the back-leak test bench—SU KWANG Precision Gold-1000 manufactured by SU KWANG Precision Co. Korea. This is an equipment developed to test whether the performance of each part of the common rail system of a vehicle in operation is operating correctly and is properly maintained. Under healthy injector conditions, a back-leak of not more than 1–2 cubic centimeters per minute (cc/min) is expected; however, upon conducting the test on the injectors after the experiment, Figure 5b,c show the back-leak results of the injectors. As shown by the back-leak results—(6, 10, 4, and 4) cc/min for injectors 1, 2, 3, and 4, respectively, poor injector performances are observed in the four injectors. This can be attributed to the water-induced wear on the injector pistons, needles, and ball seats caused by the W/D emulsions as Figures 6–8 reveal, respectively.

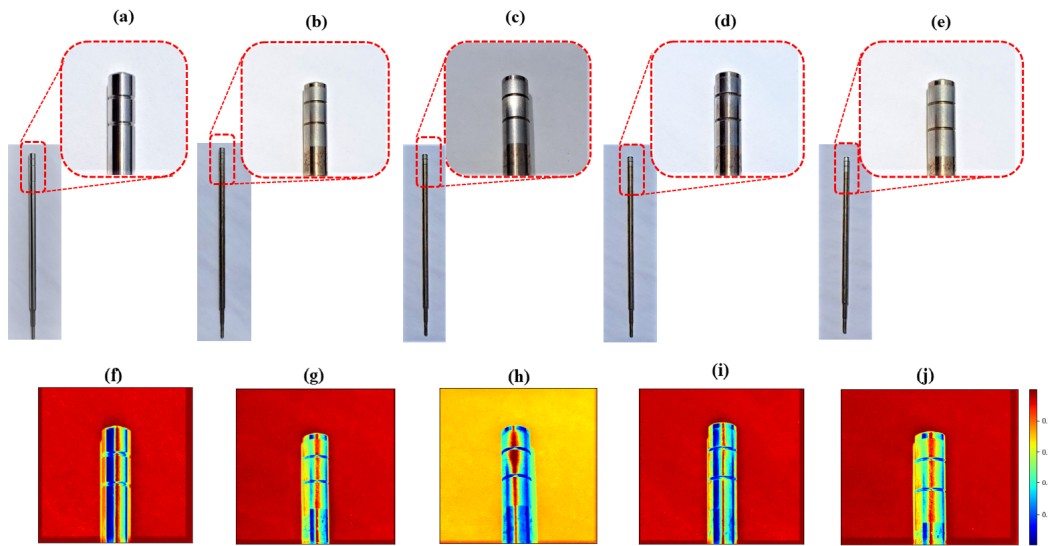

**Figure 6.** Comparison of injector valve pistons between (**a**) normal injector; (**b**–**e**) worn injectors after experiment, and (**f**–**j**) surface profiles for respective injector valve pistons.

Ideally, the valve piston slides smoothly in the valve in a reciprocating manner during operation while being lubricated by the diesel; however, as shown in Figure 6b–e, the zoomed view of the four injectors valve pistons reveal that they are worn due to friction (highlighted in red rectangles). These are in sharp contrast to a normal (new) piston as shown in Figure 6a. We suspect these wear were induced by the poor lubrication effect of the W/D emulsions used during the experiment. A closer observation of the surfaces in piston valve Figure 6g–j better reveals the wears, as shown by the high frequencies (in red and yellow) compared to that of the lower frequencies (in blue) in Figure 6f. The effects of the W/D emulsions are also observed in Figure 6b–e, which also reveal wear on the injector needles (highlighted in red rectangles) in comparison with that of a normal (new) needle as shown in Figure 6a.

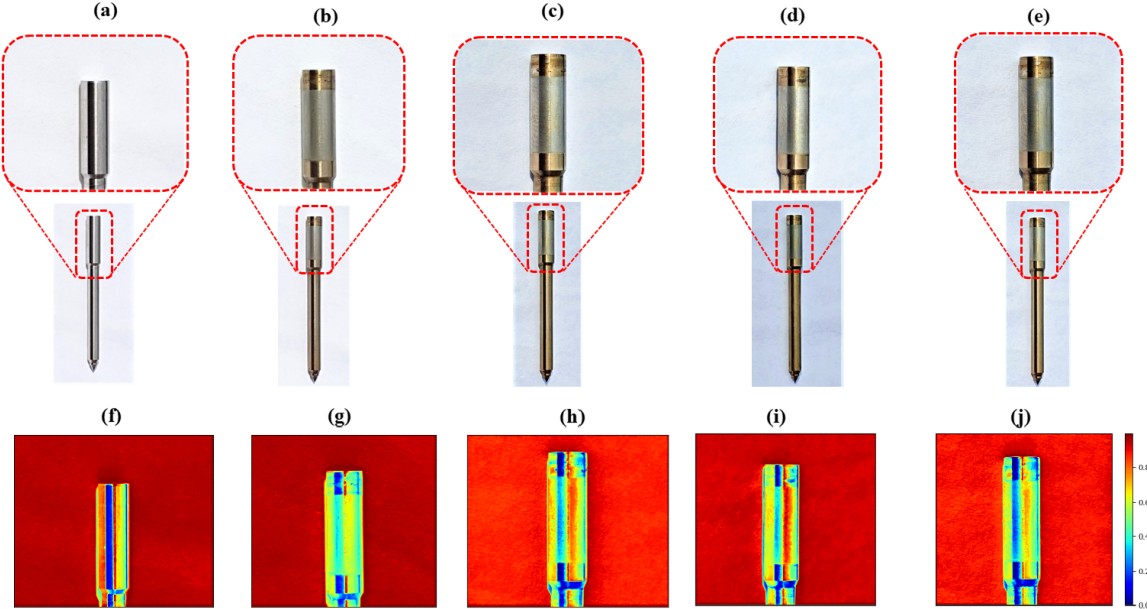

**Figure 7.** Comparison of injector nozzle needles between (**a**) normal injector, (**b**–**e**) worn injectors after experiment, and (**f**–**j**) surface profiles for respective injector valve pistons.

As shown in Figure 7b–e, wears were observed on the needle surfaces, and most likely as for the pistons, the causes for the wear are highly linked to component surface friction

caused by the poor lubrication effect of the W/D emulsion fuels. These are in contrast to the smoothness observed on the needle surface of a normal (unworn) valve piston (see Figure 7a). A closer observation of the surfaces in piston valve Figure 7g–j better reveals the wears as shown by the high frequencies (in red and yellow) compared to that of the lower frequencies (in blue) in Figure 7f.

The valve, which plays an important role among the injector components, controls the injection amount and injection timing by moving up and down of the valve ball, which maintains complete tightness with the valve seat under high pressure during nozzle opening/closing. A closer investigation for oxidation was also conducted on the valve ball seats under a magnifying lens and the results shown in Figure 8a–e.

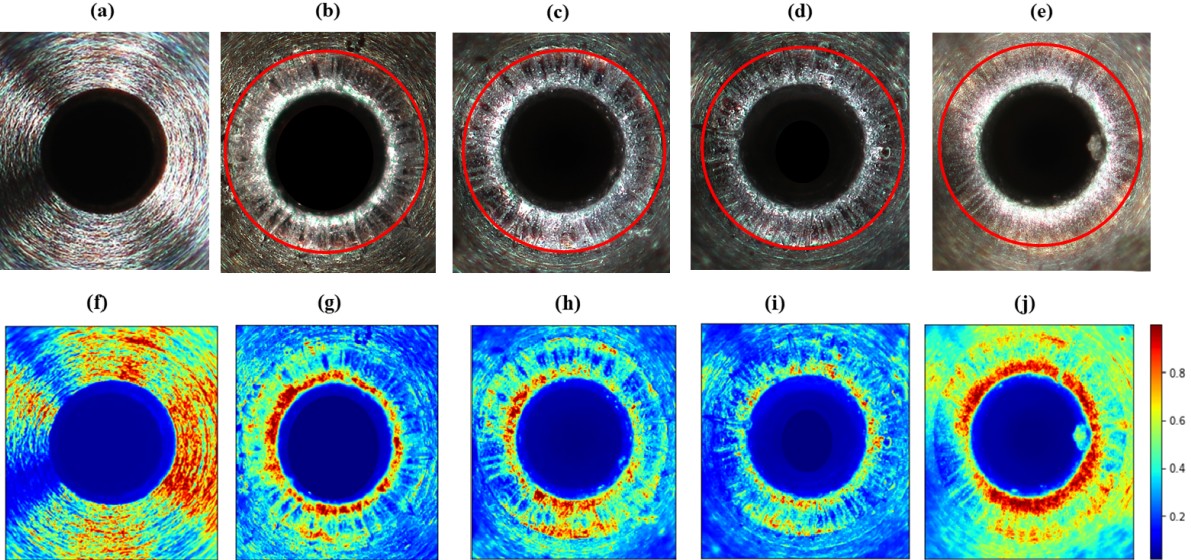

**Figure 8.** Comparison of injector valve ball seats between (**a**) normal injector, (**b–e**) worn injectors after experiment, and (**f–j**) surface profiles for respective injector valve pistons.

Tthe whiteness and corrugated circumference (highlighted in red circles) of the ball seat edges in Figure 8b–e show that some oxidation (and wear) had occurred, unlike in Figure 8a which shows a smooth circumference (and nonoxidized nature) of the ball seat edge of a normal (new) injector valve ball seat. This clearly hints at a reduction in injector efficiency. A closer observation of the surfaces in piston valve Figure 8g–j better reveals the wears as shown by the high frequencies (in red and yellow) compared to that of the lower frequencies (in blue) in Figure 8f.

### 5.2. CR System Empirical Analysis

Apart from physical observation, the digitally acquired CR pressure measurements provide reliable representative information for condition monitoring and failure assessments. At the end of the experiment, the CR pressure signals were preprocessed following a standardization technique using Equation (3) to eliminate bias and ensure the signals are respectively distributed around their mean values with unity variance. Next, the spectral condition monitoring process described in Section 3.2 was employed for investigation.

$$X' = \frac{X - \mu}{\sigma} \tag{3}$$

where $\mu$ is the mean of the CR pressure signals—$X$ while $\sigma$ is the standard deviation of $X$.

Invariably, the information provided from the FFT and PSD outputs on the respective signals reflect intuitive knowledge on the CR behavior, energy distribution, and underlying spectral characteristics. Figure 9 shows a few samples from the standardized CR signals

collected for different W/D emulsions (across the rows in unique colors) at the increasing engine speeds (down the columns)—900, 1200, 1500, 1700, and 2000 RPMs, respectively.

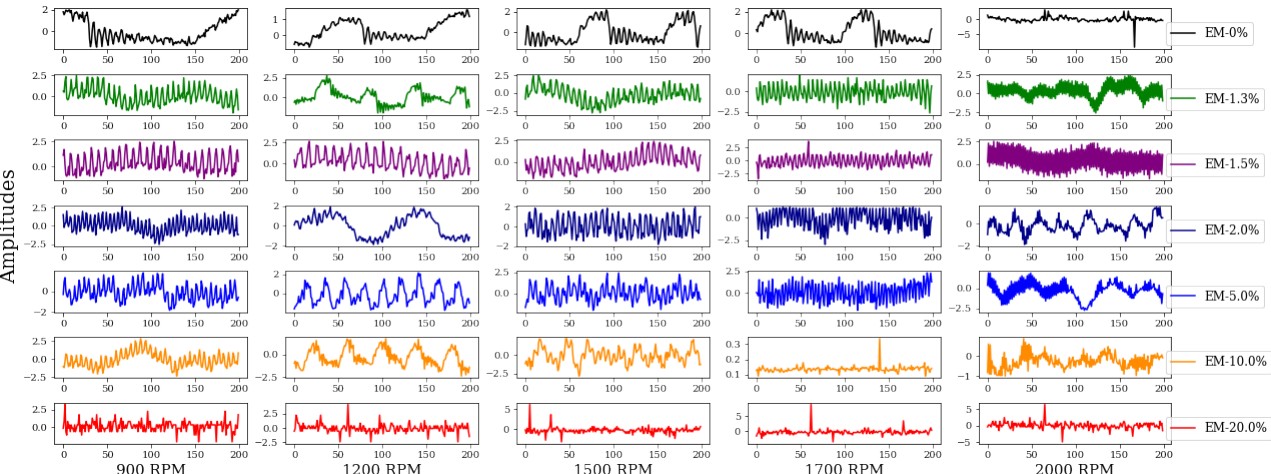

**Figure 9.** CR signals for different W/D emulsions (across the rows in unique colors) at increasing engine speeds (down columns).

As shown in the Figure 9, the transient dynamics in their respective pressure signals can be visually observed in their respective wave-forms; however, a spectral representation provides a better understanding of the CR dynamic. Figure 10 shows the FFT (on the left) and PSD (on the right) spectra comparison for the standardized CR signals collected for different W/D emulsions at the increasing engine speeds (down the columns)—900, 1200, 1500, 1700, and 2000 RPMs, respectively.

As shown in the figures, the comparison in spectra reveals the underlying CR dynamics of each W/D emulsion composition at different speeds (represented by the overlapping colors in Figure 10a–g). In Figure 10a, across the different conditions and speeds for EM-0%, the signals oscillate predominantly at lower frequencies (0–20 Hz) with several harmonics; however, it is also observed from the different spikes that as engine speed is increased, unique spectral components are observed. More obviously, in comparison between Figure 10b–g and Figure 10a, the impact of diesel emulsification on the CR system is observed from the increased spikes between 20 and 70 Hz. Notwithstanding, within this frequency range, varying engine speeds also generate several distinctive (high-energy) characteristic frequencies in the spectra. Interestingly, our suspicion of engine failure during the use of EM-20% at different engine speeds are validated by the numerous magnitudes in the FFT and PSD spectra, as shown in Figure 10g.

A closer observation of Figure 10d reveals that the lower frequency components are significantly increased when the engine speed is increased to 1500 RPM. Another increase in engine speed to 1700 RPM for the same W/D emulsion condition—EM-2.0% further agitates the 32 Hz component, and when the engine speed is further increased to 2000 RPM, new strong frequency spikes are observed around the 53 Hz in the spectra (in red). This 53 Hz spike further increases in magnitude for the succeeding condition—EM-5.0% as shown in Figure 10e, beyond which it disappears in Figure 10f—EM-10.0%. On the other hand, lower magnitudes of the lower frequency components at lower engine speeds (900–1200 RPM) are revealed alongside higher magnitudes for higher engine speeds (1500–2000 RPM). This hints at a reduced engine combustion rate (poor firing) at lower engine speeds which is intensified (accompanied with cranking sounds) as the engine speed in increased. As a result, we can hypothesize that if the water content in the W/D emulsion exceeds 10%, abnormalities may be experienced in the CR system, which may indicate a fault/failure in the injection system.

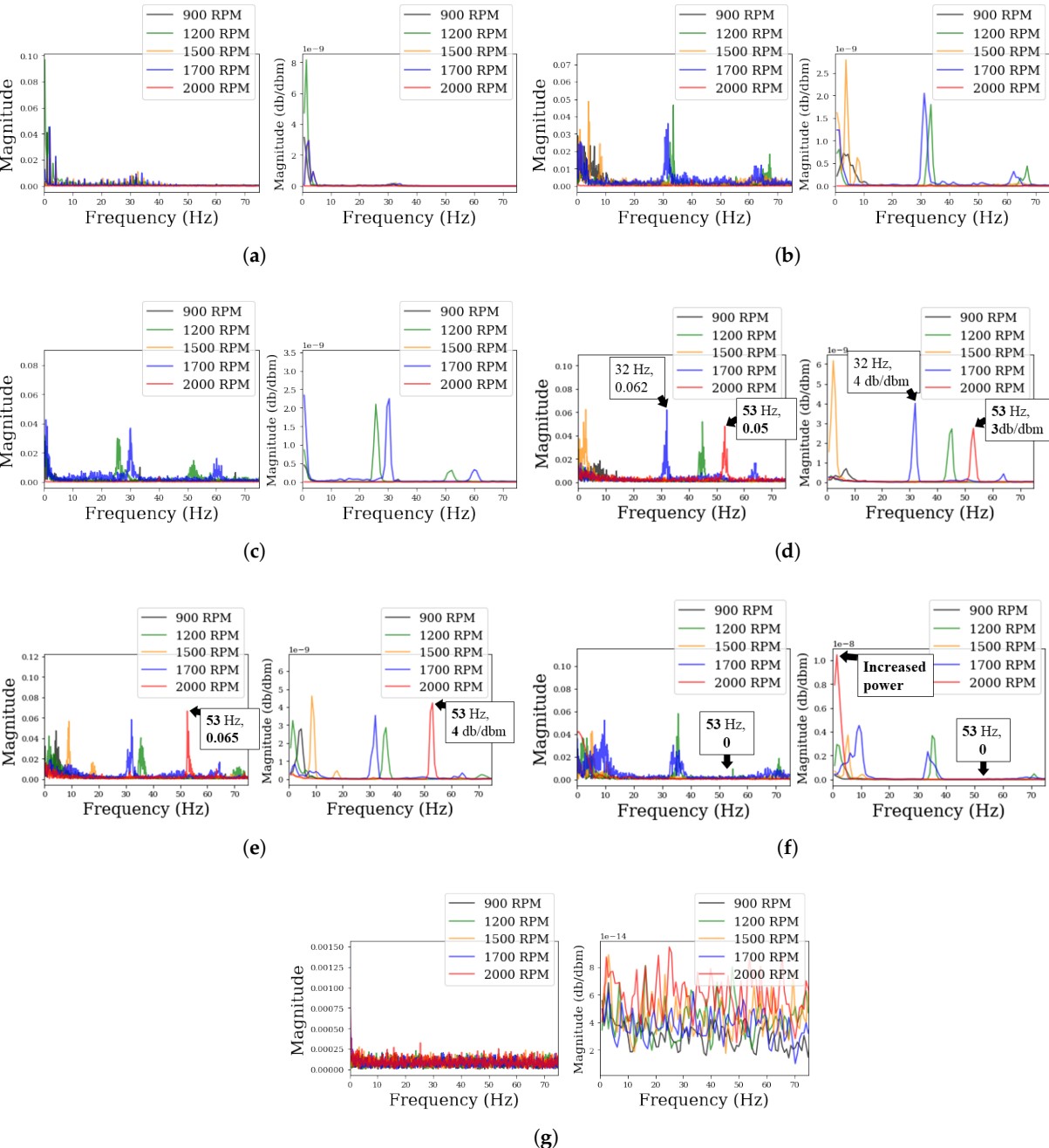

**Figure 10.** FFT (on the left) and PSD (on the right) spectra comparison of the standardized CR signals for different W/D emulsions at increasing engine speeds (**a**) EM-0%, (**b**) EM-1.3%, (**c**) EM-1.5%, (**d**) EM-2.0%, (**e**) EM-5.0%, (**f**) EM-10.0%, and (**g**) EM-20.0%.

In addition to the above results, the CR spectral behavior at the various emulsion compositions can be summarized from the spectra using several statistical parameters; from which, necessary comparisons/inferences can be made. In this study, spectral centroid ($\mu_1$), root mean square frequency (RMSF), spectral skewness (SS), and spectral kurtosis ($K_f$) were employed and are defined in Equations (4)–(7), respectively.

$$\mu_1 = \frac{\sum_{k=b_1}^{b_2} f_k s_k}{\sum_{k=b_1}^{b_2} s_k} \tag{4}$$

$$\text{RMSF} = \sqrt{\frac{\sum_{k=b_1}^{b_2} (f_k)^2}{\sum_{k=b_1}^{b_2} s_k^2}} \tag{5}$$

$$\text{SS} = \frac{\sum_{k=b_1}^{b_2} (f_k - \mu_1)^3 s_k}{(\mu_2)^3 \sum_{k=b_1}^{b_2} s_k} \tag{6}$$

$$K_f = \frac{\langle |S(t,f)|^4 \rangle}{\langle |S(t,f)|^2 \rangle^2} - 2, \quad f \neq 0 \tag{7}$$

where $f_k$ is the magnitude of bin number $k$, $s_k$ is the center frequency of $k$, $b_1, b_2$ are the band edges in bins. $\langle |S(t,f)| \rangle$ is the short-time Fourier transform (STFT) of the signals.

These parameters further reflect the CR dynamics at the different emulsion and speed conditions [27]. Spectral Centroid is a statistical value for the location of the center of mass of the spectrum that allows us to see the information on the frequency and magnitude of the spectral transformation [28], while RMSF reflects the overall energy level across the spectra. Spectral kurtosis reveals how the impulsiveness of the CR pressure signals vary with frequency, while spectral skewness measures the symmetry of the spectrum around its arithmetic mean. Ideally, the skewness return a zero value for a normal distribution and high (positive) values for signals with substantial energy. Figure 11a–d show the comparison using these parameters for different emulsion compositions at increasing engine speeds.

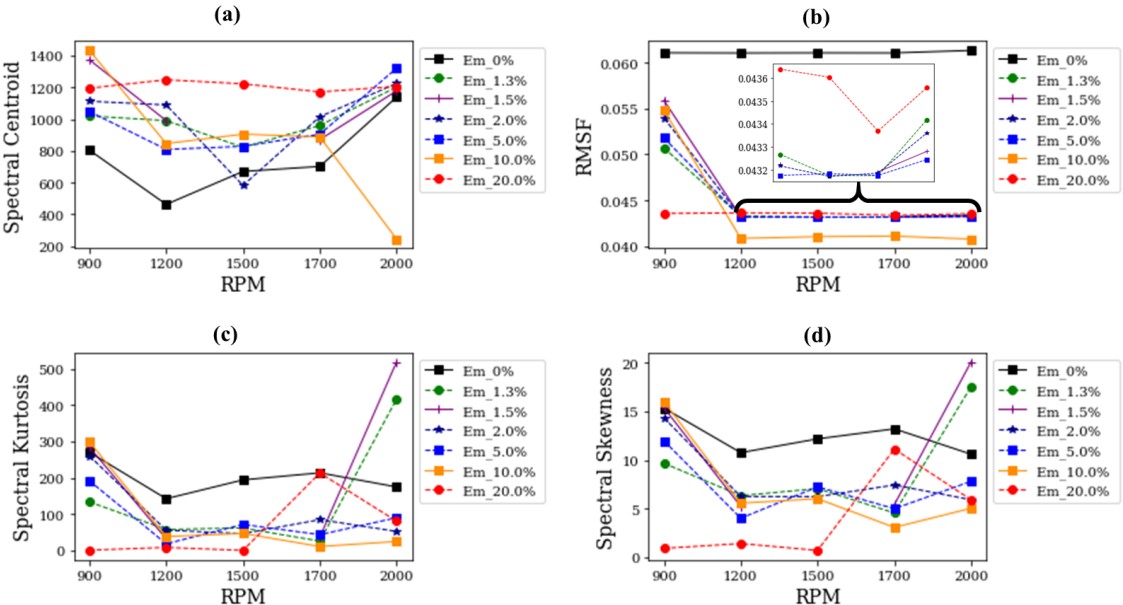

**Figure 11.** Spectral parameters of CR pressure differentials for different emulsion compositions (**a**) spectral Centroid, (**b**) RMSF, (**c**) Spectral kurtosis, and (**d**) Spectral skewness.

It is observed in Figure 11a that the spectral centroid at EM-0% and EM-10%, are lowest and highest, respectively, at engine idling condition (900 RPM). This validates the contrasting health conditions between the use of pure diesel fuel and the use of W/D fuels. Regardless of the increasing CR vigorosness caused by W/D emulsions, the spectral centroid values for EM-1.3%–EM-2.0% seem to be within acceptable ranges for lower engine speeds (900–1500 RPM). Also from Figure 11b, the CR energy levels are generally reduced by increasing W/D emulsions at engine idling condition (900 RPM). Increasing the engine speed at these conditions doesn't have a significant impact on the energy levels, except for high emulsion compositions—EM-10.0% (and above), where the energy levels are significantly further decreased at increased engine speeds. The spectral kurtosis values

in Figure 11c reveal a general decrease in impulsiveness as the water content is increased; however, as engine speed is increased, the impulsiveness is generally observed to increase. In addition, Figure 11d suggests that small emulsion compositions (EM-1.3%–EM-5.0%) at engine speeds which produce car mobility for safe-zone driving (between 1200 and 1700 RPM of engine speed) produce pressure signals close to using clean diesel fuel (EM-0%); unfortunately, at higher engine RPMs (2000 RPM), the correlation between the signal shapes at increasing water-emulsion composition (and increasing speeds) is unclear.

The results reported so far reveal the characteristics and damage severity of a CRD engine FIS under varying W/D emulsion compositions using statistical and exploratory methods. This is in line with evaluating the potentials of diesel emulsification for commercial use towards promoting the campaign for greener energy and emission reduction. Popular knowledge supports the fact that diesel emulsification improves diesel quality and helps reduce exhaust temperature (as is shown herein); however, the impeding (long-term) effects on critical injection components as this study reveals is a factor worth considering before adoption. On a different note, since modern automotive design technologies are aimed at not just production cost and emission reduction but at achieving higher power densities for mass-produced automotive diesel engines [12–14], it may be an uphill task to achieve a global paradigm for optimally achieving these feats in one single cost-efficient diesel engine since the realization that one may imply forfeiting the other(s). For instance, even with engine downsizing as a promising solution [10,11], realizing advanced FIS capable of very high injection pressures (2700 bar and above) without wear issues may be impeded by diesel emulsification. While their simulated investigations on a high-power design- ultralight midsize diesel engine matches future $CO_2$ targets [14], concerns for real-life applicability; especially with diesel emulsification are still open for continued investigations.

With the already-increasing campaign to reduce (rather than maintain) engine emission levels on a global scale, additional concerns on the wear effects of W/D emulsion fuels in such high-power designs are still open for further investigations. Although in small W/D compositions, the effect of W/D fuels on engine efficiency may be marginal but most probably, with large W/D compositions, the efficiency of such engines (power output) may be threatened in short-term. Nonetheless, regardless of the W/D emulsion level, the long-term wear effect on the injector components seems inevitable. Clearly, fuel quality is fundamental for FIS component durability and should be prioritized for overall engine durability. This presents an open ground for continued investigations towards the development/discovery of oxidation-resistant materials for FIS design.

## 6. Concluding Remarks

A closer look into water emulsified diesel fuels reveals that diesel chemical property is improved by emulsification, and this can be attributed to the phenomena—*micro explosions*. Although existing studies reveal encouraging paradigms for the use of W/D emulsion fuels for improved fuel efficiency and a more ecofriendly environment, the reliability of such methods remains open for continued discussions. To analyze the effect of fuel mixed with water on the diesel engine injection system, this study uses a KIA Sorento 2004 four-cylinder line engine as a case study and investigated the CRD injection system using visual and spectral analysis. Results reveal the wear effects of W/D emulsions on injection system components and deliver the following conclusions:

- We can grasp the behavior of the injection system through FFT and PSD analysis of the CR pressure signals;
- If the water content exceeds 10% or the engine speed exceeds 1500 RPM, it may have a fatal adverse effect on the injection system;
- The amount of back-leak of the injectors was found to be (4 to 10) *cc*, which is a state in which repair and inspection are required due to the deterioration of the injector's performance;

- The cause of the failure of the injection system was confirmed to be the wear of the valve piston, nozzle needle, and ball seat of the injector;
- Wear of injector parts due to emulsified fuel with water is projected to result from metal oxidation, cavitation, and reduction in lubricity.

So far, the empirical assessments discussed herein apply only to a single case study under passive control of operating conditions—the W/D emulsion components were pre-designed, and several additional tests on different engine models (and configuration) may be needed for a more comprehensive deduction on the damage severity of FIS components from diesel emulsification, even with emission control efficiencies.

**Author Contributions:** Conceptualization, M.-S.K., U.E.A. and J.-W.H.; methodology, M.-S.K. and U.E.A.; software, U.E.A.; formal analysis, U.E.A.; investigation, M.-S.K. and U.E.A.; resources, M.-S.K., U.E.A. and J.-W.H.; data curation, M.-S.K. and U.E.A.; writing-original draft—M.-S.K., writing—review and editing, U.E.A. and J.-W.H.; visualization, U.E.A.; supervision, J.-W.H.; project administration, J.-W.H.; funding acquisition, J.-W.H. All authors have read and agreed to the published version of the manuscript.

**Funding:** This work was supported by the National Research Foundation of Korea (NRF) grant funded by the Korea government (MIST) (No. 2019R1/1A3A01063935).

**Institutional Review Board Statement:** Not applicable.

**Informed Consent Statement:** Not applicable.

**Data Availability Statement:** The data presented in this study are available on request from the corresponding author. The data are not publicly available due to laboratory regulations.

**Conflicts of Interest:** The authors declare no conflict of interest.

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
