# Peer review of "A Study on Water-Induced Damage Severity on Diesel Engine Injection System Using Emulsified Diesel Fuels"

_electronics, doi:10.3390/electronics10182285_

Round 1

Reviewer 1 Report

Grammar error in abstract. a host many problems

Please add brief specifications of transducers and DAQ device. Please check the dynamic frequency response range of the pressure sensor.

Line 161. Please make the legend more logical. It is not easy to follow now.

Figure 7 and 8. It is not easy to compare the wear degree. Please measure surface profiles.

The spectrum analysis of the CR pressure signals should show large peaks (0.5*fr for a four-stroke engine and harmonics) in a low frequency range regarding to the fuel pumping or injection. Please consider increase the length of samples, which can increase the frequency resolution.

Please explain the peak in the high frequency, for instance about 1000Hz in Figure 10. Reasonably the frequency is relating to the engine operation. The frequency varies from EM fuels and please make sure the running speeds are close between tests.

Regarding to the results in Figure 14, a higher sampling frequency is recommended.

It would be better if the spectra at same speeds have the identical limits for comparison. Alternatively the authors may give an overlapped figure of spectra.

Figure 15. The results may be improved if some informative frequency bands are selected other than the whole frequency band.

The wear mechanism is not sufficiently addressed. The cavitation could be the major factor. I am not expecting the authors explain thoroughly in this manuscript. It would be very interesting for the future work.

Reviewer 2 Report

-The topic is interesting, but significant modifications are required.

-The abstract section is not sufficiently developed in terms of background, originality method and qualitative/quantitative results. Then some parts related to world energy demand is not well linked to the topic.

-The introduction section should be revised to improve all parts, as is required by a high quality paper. Also, in this section the originality of the work conducted is not clear. The scientific contribution should also be clearly stated. Refer also to very advanced CR injection systems capable of very high-injection pressures, for which fuel quality is fundamental: 10.4271/03-12-02-0010, 10.4271/2017-24-0072, 10.1016/j.fuel.2017.06.112, 

-Fig 1 and 2 are not fundamental, they are just examples. Provide sketches or schemes of your own system.

-Fig 4. add  y-unit as deg C, improve identification of rpm on x-axis.

-Then improve the discussion section in terms of scientific sound in relation to the results observed. Then add notes on what can be the effect on the newest FIS capable of up to 2700 bar and what the authors expect on those systems.

The conclusion section should be more effective, but overall good structured.

Overall: the global quality should improve from the scientific perspective, but the topic and the method are correct and interesting. Addressing all reviewer comments should permit to improve the presentation and the quality sufficiently to be approved for publishing.

Thanks

Round 2

Reviewer 1 Report

The response time (≤5ms) is not fast enough and the sensor is not able to capture the high frequency responses. The high frequency characteristics may be not accurate.

The authors have clearly explained the other questions.

Reviewer 2 Report

The authors have sufficiently addressed reviewer comments. No additional modifications are requested.

Thanks

Author Response

We are glad that our manuscript interests the reviewer